# Learning to Constrain Policy Optimization with Virtual Trust Region

**Hung Le, Thommen Karimpanal George, Majid Abdolshah, Dung Nguyen,**
**Kien Do, Sunil Gupta, Svetha Venkatesh**
Applied AI Institute, Deakin University, Geelong, Australia
`thai.le@deakin.edu.au`

## Abstract

We introduce a constrained optimization method for policy gradient reinforcement learning, which uses a *virtual trust region* to regulate each policy update. In addition to using the proximity of one single old policy as the normal trust region, we propose forming a second trust region through another *virtual policy* representing a wide range of past policies. We then enforce the new policy to stay closer to the virtual policy, which is beneficial if the old policy performs poorly. More importantly, we propose a mechanism to automatically build the virtual policy from *a memory of past policies*, providing a new capability for dynamically learning appropriate virtual trust regions during the optimization process. Our proposed method, dubbed Memory-Constrained Policy Optimization (MCPO), is examined in diverse environments, including robotic locomotion control, navigation with sparse rewards and Atari games, consistently demonstrating competitive performance against recent on-policy constrained policy gradient methods.

## 1 Introduction

Deep reinforcement learning (RL) is the current workhorse in machine learning. Using neural networks to approximate value and policy functions enables classical approaches such as Q-learning [29] and policy gradient [23] to achieve promising results on many challenging problems such as Go, Atari games and robotics [21, 15, 12, 13]. Compared to Deep Q-learning, deep policy gradient (PG) methods are often more flexible and applicable to discrete and continuous action problems. However, these methods tend to suffer from high sample complexity and training instability since the gradient may not accurately reflect the policy gain when the policy changes substantially [6]. This is exacerbated for deep policy networks where numerous parameters need to be optimized, and minor updates in parameter space can lead to considerable changes in policy space.

To address this issue, one solution is to regularize each policy update by restricting the Kullback–Leibler (KL) divergence between the new policy and the previous one, which can guarantee monotonic policy improvement [17]. However, jointly optimizing the approximate advantage function and the KL term does not work in practice [17]. Therefore, Schulman et al. (2015) proposed Trust Region Policy Optimization (TRPO) to constrain the new policy within a KL divergence radius, which requires second-order gradients. Alternatives such as Proximal Policy Optimization (PPO) [19] use a simpler first-order optimization with adaptive KL or clipped surrogate objective while still maintaining the reliable performance of TRPO. Recent methods recast the problem through a new lens using Expectation-Maximization or Mirror Descent Optimization, and this also results in first-order optimization with KL divergence term in the loss function [1, 22, 31, 24].

An issue with the above methods is that the previous policy used to restrict the new policy may be suboptimal and thus unreliable in practice. For example, due to stochasticity and approximations, the new policy may fall into a local optimum even under trust-region optimizations. Then in the

next update, this policy will become the "previous" policy, and will continue pulling the next policy to stay in the local optimum, thus slowing down the training progress. For on-policy methods using mini-batch updates like PPO, the situation is more problematic as the "previous" policy is defined as the old policy to collect data, which can be either very far or close to the current policy. There is no guarantee that the old policy defines a reasonable trust region for regulating the new policy.

In this paper, we propose a novel constrained policy iteration procedure, dubbed Memory-Constrained Policy Optimization (MCPO), wherein a *virtual policy* representing memory of past policies regularizes each policy update. The virtual policy forms a virtual trust region, attracting the new policy more when the old policy performs badly, which prevents the optimization from falling into local optimum caused by the old policy of poor quality. As such, we measure 2 KL divergences corresponding to the virtual and old policy, and assign different weights to the two KL terms in building the objective function. The weights are computed dynamically based on the performance of the two policies (the higher performer yields higher weights).

In contrast to prior works using heuristics (e.g. running average or mean of past policies) to form additional trust regions [28], we argue that the virtual policy should be determined dynamically to maximize the performance on current training data. Thus we store policies in a policy memory, and learn to extract the most relevant ones to the current context. We sum the past policies in a weighted manner wherein the weights are generated by a neural network–named the *attention network*, which takes the information of the current, the old and the last virtual policy as the input. The attention network is optimized to maximize the approximate expected advantages of the virtual policy. We jointly optimize the policy and attention networks to train our system, alternating between sampling data from the policy and updating the networks in a mini-batch manner.

We verify our proposed MCPO through a diverse set of experiments and compare our performance with that of recent constrained policy optimization baselines. In our experiment on classical control tasks, amongst tested models, MCPO consistently achieves better performance across tasks and hyperparameters. Our testbed on 6 Mujoco tasks shows that MCPO with a big policy memory is performant where the attention network plays an important role. We also demonstrate MCPO's capability of learning efficiently on sparse reward and high-dimensional problems such as navigation and Atari games. Finally, our ablation study highlights the necessity of MCPO's components such as the virtual policy and the attention network.

## 2 Background: Policy Optimization with Trust Region

In this section, we briefly review some fundamental constrained policy optimization approaches. A general idea is to force the new policy $\pi_\theta$ to be close to a recent policy $\pi_{\theta_{old}}$. In this paper, we usually refer to a policy via its parameters (i.e. policy $\theta$ means policy $\pi_\theta$). We also use finite-horizon estimators for the advantage with discount factor $\gamma \in (0, 1)$ and horizon $T$.

**Conservative Policy Iteration (CPI)** This method starts with a basic objective of policy gradient algorithms, which is to maximize the expected advantage $\hat{A}_t$.

$$L^{CPI}(\theta) = \hat{\mathbb{E}}_t \left[ \frac{\pi_\theta(a_t|s_t)}{\pi_{\theta_{old}}(a_t|s_t)} \hat{A}_t \right]$$

where the advantage $\hat{A}_t$ is a function of returns collected from $(s_t, a_t)$ by using $\pi_{\theta_{old}}$ (see Appendix A.2) and $\hat{\mathbb{E}}_t[\cdot]$ indicates the empirical average over a finite batch of data. To constrain policy updates, the new policy is a mixture of the old and the greedy policy: $\tilde{\theta} = \operatorname{argmax} L^{CPI}(\theta)$. That is, $\theta = \alpha\theta_{old} + (1 - \alpha)\tilde{\theta}$ where $\alpha$ is the mixture hyperparameter [6]. As the data is sampled from the previous iteration's policy $\theta_{old}$, the objective needs importance sampling estimation. Hereafter, we denote $\frac{\pi_\theta(a_t|s_t)}{\pi_{\theta_{old}}(a_t|s_t)}$ as $\tau_t(\theta)$ for short.

**KL-Regularized Policy Optimization** To enforce the constraint, one can jointly maximize the advantage and minimize KL divergence between the new and old policy, which ensures monotonic improvement [17].

$$L^{KL}(\theta) = \hat{\mathbb{E}}_t \left[ \tau_t(\theta) \hat{A}_t - \beta KL \left[ \pi_{\theta_{old}}(\cdot|s_t), \pi_\theta(\cdot|s_t) \right] \right]$$

where $\beta$ is a hyperparameter that controls the update conservativeness, which can be fixed (KL Fixed) or changed (KL Adaptive) during training [19].

**Trust Region Policy Optimization (TRPO)** The method optimizes the expected advantage with hard constraint [17]. This is claimed as a practical implementation, less conservative than the theoretically justified algorithm using KL regularizer mentioned above.

$$L^{TRPO}(\theta) = \hat{\mathbb{E}}_t \left[ \tau_t(\theta) \hat{A}_t \right]$$
$$\text{st} \ \delta \geq KL\left[ \pi_{\theta_{old}}(\cdot|s_t), \pi_\theta(\cdot|s_t) \right]$$

where $\delta$ is the KL constraint radius.

**Proximal Policy Optimization (PPO)** PPO is a family of constrained policy optimization, which uses first-order optimization and mini-batch updates including KL Adaptive and clipped PPO. In this paper, we use PPO to refer to the method that limits the change in policy by clipping the loss function (clipped PPO) [19]. The objective $L^{PPO}$ is defined as

$$\hat{\mathbb{E}}_t \left[ \min \left( \tau_t(\theta) \hat{A}_t, \text{clip}(\tau_t(\theta), 1-\epsilon, 1+\epsilon) \hat{A}_t \right) \right]$$

where $\epsilon$ is the clip hyperparameter.

In all the above methods, $\theta$ is the currently optimized policy, which is also referred to as the current policy. $\theta_{old}$ represents a past policy, which can be one or many update steps before the current policy. In either case, the rule to decide $\theta_{old}$ is fixed throughout training. If $\theta_{old}$ is suboptimal, it is unavoidable that the following updates will be negatively impacted. We will address this issue in the next section.

## 3 Memory-Constrained Policy Optimization

In trust-region policy gradient methods with mini-batch updates such as PPO, the old policy $\theta_{old}$ is often the last *"sampling"* policy for collecting observations from the environment. This policy is, by design, fixed during updates of the main policy until the next interaction with the environment. This means $\theta_{old}$ may not immediately precede the current policy $\theta$ but may be from many (update) steps before. Since $\theta_{old}$ is not up-to-date and could possibly be not good, using it to constrain the current policy $\theta$ can cause the suboptimal update of $\theta$. Similarly, the immediately preceding policy could also be poor in quality and thus, using it as $\theta_{old}$ could be detrimental to the optimization.

To tackle this issue, we propose to learn to constrain $\theta$ towards a *weighted combination of multiple past policies* besides $\theta_{old}$. We argue that we can optimize the attention network to ensure the combined policy is optimal. Below is formal description of our method.

### 3.1 Virtual Policy

**Computing the virtual policy via past policies** The virtual policy should be determined based on the past policies and their contexts such as quality, distance or entropy. A simple strategy such as taking average of past policies is likely suboptimal as the quality of these policies vary and some can be irrelevant to the current learning context. Let $\psi$ be the weighted combination of $M$ past policies $\{\theta_i\}_{i=1}^M$, which we refer to as a *"virtual"* policy for naming convenience, $\psi$ is computed as follows:

$$\psi = \sum_{i=1}^{M} f_\varphi(v)_i \theta_i \tag{1}$$

where $f_\varphi$ is a neural network parameterized by $\varphi$ which outputs softmax attention weights over the $M$ past policies, $v$ is a *"context"* vector capturing different relations among the current policy $\theta$, the last sampling policy $\theta_{old}$, and the last virtual policy $\psi_{old}$. We build the context by extracting specific features: pair-wise distances between policies, the empirical returns of these policies, policy entropy and value losses (details in Appendix Table 4). Intuitively, these features suggest which virtual policy will yield high performance (e.g., a virtual policy that is closer to the policy that obtained high return and low value loss). The details of $f_\varphi$ training will be given in Sec. 3.2.

**Algorithm 1** Memory-Constrained Policy Optimization.

---

**Require:** A policy buffer $\mathcal{M}$, an initial policy $\pi_{\theta_{old}}$. $T$, $K$, $B$ are the learning horizon, number of update epochs, and batch size, respectively.
1: Initialize $\psi_{old} \leftarrow \theta_{old}$, $\theta \leftarrow \theta_{old}$
2: **for** $iteration = 1, 2, ...$ **do**
3:     Run policy $\pi_{\theta_{old}}$ in environment for $T$ timesteps. Compute advantage estimates $\hat{A}_1, ..., \hat{A}_T$
4:     **for** $epoch = 1, 2, ...K$ **do**
5:         **for** $batch = 1, 2, ...T/B$ **do**
6:             Compute $\psi$ (Eq. 1) using $\psi_{old}$, $\theta$, $\theta_{old}$, optimize $\theta$ and $\varphi$ by maximizing $L^{MCPO}$ (Eq. 6)
7:             **if** $D(\theta, \psi) > D(\theta_{old}, \psi)$ **then** add $\theta$ to $\mathcal{M}$
8:             **if** $|\mathcal{M}| > N$ **then** remove the oldest item in $\mathcal{M}$
9:             $\psi_{old} \leftarrow \psi$
10:         **end for**
11:     **end for**
12:     $\theta_{old} \leftarrow \theta$
13: **end for**

---

**Storing past policies with diversity-promoting writing**    We use a memory buffer $\mathcal{M}$ to store past policies. We treat $\mathcal{M}$ as a *queue* with maximum capacity $N$, which means a new policy $\theta$ will be added to the end of $\mathcal{M}$ and if $\mathcal{M}$ is full, the oldest policy will be discarded. However, we do *not* add any new policy $\theta$ to $\mathcal{M}$ unconditionally but only when $\theta$ satisfies our *"diversity-promoting"* condition. Let $D(a, b) = \hat{\mathbb{E}}_t [KL [\pi_a (\cdot|s_t), \pi_b (\cdot|s_t)]]$ denote the "distance" between 2 policies $\pi_a$ and $\pi_b$, the *diversity-promoting* writing is defined as:

$$\text{Adding } \theta \text{ to } \mathcal{M} \text{ if } D(\theta, \psi) \geq D(\theta_{old}, \psi) \tag{2}$$

where $D(\theta_{old}, \psi)$ serves as a threshold. This condition makes sure that the policy to be added is far enough from $\psi$. It is reasonable because if $\theta$ is too similar to $\psi$ (in regard of $\theta_{old}$), the advantage of storing multiples past policies in order to find a good one will disappear. We will elaborate more on this in Sec. 4.5.

## 3.2   Policy Optimization with Two Trust Regions

**Optimizing the policy parameters** $\theta$    During policy optimization, we make use of both $\theta_{old}$ and $\psi$ to constrain $\theta$. Hence, $\theta_{old}$ and $\psi$ form 2 trust regions, and we aim to enforce the new policy to be closer to the better one. To this end, we propose to learn a new policy $\theta$ by maximizing the following objective function:

$$\begin{aligned}
L_1(\theta) =& \hat{\mathbb{E}}_t \left[ \tau_t(\theta)\hat{A}_t \right] \\
& - \beta\hat{\mathbb{E}}_t \bigg[ (1 - \alpha_t (\cdot|s_t)) KL [\pi_{\theta_{old}} (\cdot|s_t), \pi_\theta (\cdot|s_t)] \\
& + \alpha_t (\cdot|s_t) KL [\pi_\psi (\cdot|s_t), \pi_\theta (\cdot|s_t)] \bigg]
\end{aligned} \tag{3}$$

where $\beta$ is the weight balancing between the main objective and the policy constraints, $\alpha_t$ is the weight balancing between constraining $\theta$ towards $\psi$ and $\theta$ towards $\theta_{old}$. The expectation is estimated by taking average over $t$ in a mini-batch of sampled data. We note that in the early stage of learning, the attention network is not trained well, and thus the quality of the virtual policy $\psi$ may be worse than $\theta_{old}$. In the long-term, $\psi$ will get better and tend to provide a better trust-region. Hence, we need to use both trust regions to ensure maximal performance at any learning stage. We can also prove that using our proposed two trust regions guarantees monotonic policy improvement (see Appendix. C). Below we introduce a mechanism to automatically determine the contribution of the two trust regions through computing $\alpha$ and $\beta$ coefficients.

Intuitively, if the virtual policy is better than the old policy, the new policy should be kept close to the virtual policy and vice versa. Hence, $\alpha_t$ should be proportional to the contribution of $\psi$ to the final performance with regard to that of $\theta_{old}$. Thus, we define: $\alpha_t (\cdot|s_t) = \frac{\exp(R_t(\psi))}{\exp(R_t(\psi)) + \exp(R_t(\theta_{old}))}$

| Model | Pendulum 1M | LunarLander 1M | BWalker 5M |
|---|---|---|---|
| KL Adaptive ($d_{targ} = 0.01$) | -147.52±9.90 | *254.26±19.43* | 247.70±14.16 |
| KL Fixed ($\beta = 0.1$) | -464.29±426.27 | *256.75±20.53* | *263.56±10.04* |
| PPO (clip $\epsilon = 0.3$) | -591.31±229.32 | *259.93±22.52* | *260.51±17.86* |
| MDPO ($\beta_0 = 2$) | *-135.52±5.28* | 227.76±16.96 | 226.80±15.67 |
| VMPO ($\alpha_0 = 1$) | *-139.50±5.54* | 212.85±43.35 | 238.82±11.11 |
| MCPO ($N = 5$) | **-133.42±4.53** | *262.23±12.47* | *265.80±5.55* |
| MCPO ($N = 10$) | -146.88±3.78 | *263.04±11.48* | **266.26±8.87** |
| MCPO ($N = 40$) | *-135.57±5.22* | **267.19±13.42** | 249.51±12.75 |

Table 1: Mean and std. over 5 runs on classical control tasks (with number of training steps). Bold denotes the best mean. Underline denotes good results (if exist), statistically indifferent from the best in terms of Cohen effect size less than 0.5. The baselines are reported with best hyperparameters.

where $R_t(\psi)$, $R_t(\theta_{old})$ are the *estimated returns* corresponding to $\psi$ and $\theta_{old}$, respectively. We estimate $R_t(\cdot)$ via weighted importance sampling, that is, $R_t(\cdot) = \tau_t(\cdot)\hat{A}_t$ where both are computed using the same $s_t$. We also dynamically adjust $\beta$ by switching between 2 values $\beta_{min}$ and $\beta_{max}$ ($0 < \beta_{min} < \beta_{max}$) as follows:

$$\beta = \begin{cases} \beta_{max} & \text{if } D(\theta_{old}, \theta) > D(\theta_{old}, \psi) \\ \beta_{min} & \text{otherwise} \end{cases} \tag{4}$$

where $D(\cdot, \cdot)$ is again the KL "distance". The reason behind this update of $\beta$ is to encourage stronger enforcement of the constraint when $\theta$ is too far from $\theta_{old}$. Unlike using a fixed threshold $d_{targ}$ to change $\beta$ (e.g in KL Adaptive, if $D(\theta_{old}, \theta) > d_{targ}$, increase $\beta$ [19]), we make use of $\psi$ as a reference for selecting $\beta$. This allows a dynamic threshold that varies depending on the current learning. We name this mechanism as *switching-$\beta$* rule.

**Optimizing the attention network parameters** $\varphi$ To encourage the attention network $f_\varphi$ (Eq. 1) to produce the optimal attention weights over past policies, we maximize the following objective:

$$L_2(\varphi) = \hat{\mathbb{E}}_t \left[ R_t(\psi_\varphi) \right] \tag{5}$$

which is the approximate expected return w.r.t. $\psi_\varphi$. We write $\psi_\varphi$ to emphasize that $\psi$ is a function of $\varphi$ (as formulated in Eq. 1). We use gradient ascent to update $\varphi$ by backpropagating the gradient $\frac{\partial \mathcal{L}_2}{\partial \varphi}$ through the attention network.

By learning $f_\varphi$, we can guarantee that the virtual policy $\psi$ is the best combination of past policies in terms of return. Note that for on-policy learning setting, optimizing "soft" attention weights is usually more robust than searching for the best past policy in $\mathcal{M}$ because the estimation of the expected return using current data samples can be noisy and is not always reliable. Also, searching is computationally more expensive than using the attention.

**Final Objective** We train the whole system by maximizing the following objective:

$$L^{MCPO} = L_1(\theta) + L_2(\varphi) \tag{6}$$

where MCPO stands for *Memory-Constrained Policy Optimization*. We optimize $\theta$ and $\psi$ alternately by fixing one and learning the other. We implement MCPO using minibatch update procedure [19]. MCPO Pseudocode is given in in Algo 1. For notational simplification, the algorithm uses 1 actor.

## 4 Experimental results

In our experiments, we show our optimization scheme is superior in different aspects: hyperparameter sensitivity, sample efficiency and consistent performance in continuous and discrete action spaces. The main baselines are recent on-policy constrained methods that use first-order optimization, in which most of them employ KL terms in the objective function. They are KL Adaptive, KL Fixed, PPO [19], MDPO [24], VMPO [22] and TRGPPO [27] . We also include second-order

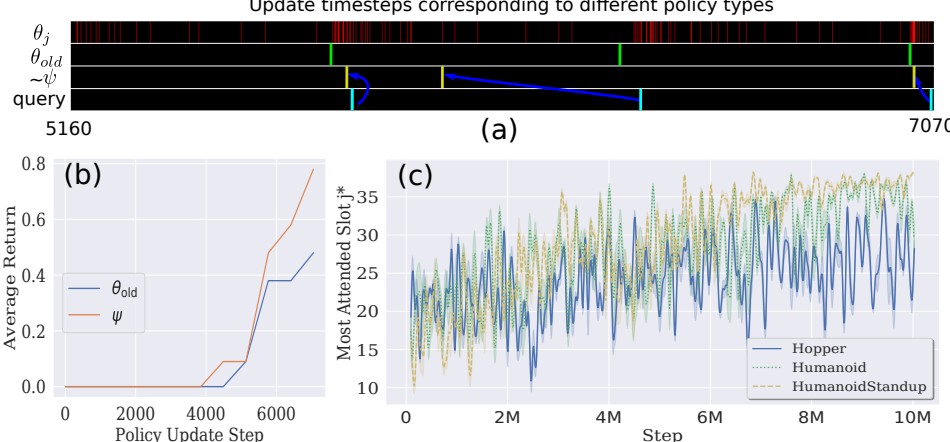

Figure 1: (a) Policy analysis on Unlock. First row (red lines): steps where a policy is added to $\mathcal{M}$, i.e. the steps of $\theta_j$. Second row (green lines): steps of old policies $\theta_{old}$. Third row (yellow lines): steps of mostly attended policy, approximating $\psi$. Fourth row (cyan lines): 3 steps of interest where we want to find their attended steps. Blue arrows link a query step and the step that receives highest attention. (b) Quality of $\psi$ vs. $\theta_{old}$. Average return collected by $\psi$ and $\theta_{old}$ at different stages of training. (c) 3 Mujoco tasks. The slot in $\mathcal{M}$ received the highest attention $j^* = \arg\max_j f_\varphi (v_{context})_j$ over time.

methods such as TRPO [17] and ACKTR [30]. Across experiments, for MCPO, we fix $\beta_{max} = 10$, $\beta_{min} = 0.01$ and only tune $N$. More details on the baselines and tasks are given in Appendix B.1. Our code is available at `https://github.com/thaihungle/MCPO`.

## 4.1 Classical Control

In this section, we compare MCPO to other first-order policy gradient methods (KL Adaptive, KL Fixed, PPO, MDPO and VMPO) on 3 classical control tasks: Pendulum, LunarLander and Bipedal-Walker, which are trained for one, one and five million environment steps, respectively. Here, we are curious to know how the model performance fluctuates as the hyperparameters vary. For each model, we choose one hyperparameter that controls the conservativeness of the policy update, and we try different values for the signature hyperparameter while keeping the others the same. For example, for PPO, we tune the clip value $\epsilon$; for KL Fixed we tune $\beta$ coefficient and these possible values are chosen following prior works. For our MCPO, we tune the size of the policy memory $N$ (5, 10 and 40). We do not try bigger policy memory size to keep MCPO running efficiently (see Appendix B.2 for details).

Table 1 reports the results of MCPO and 5 baselines with the best hyperparameters. For these simple tasks, tuning the hyperparameters often helps the model achieve at least moderate performance. However, models like KL Adaptive and VMPO cannot reach good performance despite being tuned. PPO shows good results on LunarLander and BipedalWalker, yet underperforms others on Pendulum. Interestingly, if tuned properly, the vanilla KL Fixed can show competitive results compared to PPO and MDPO in BipedalWalker. Amongst all, our MCPO with suitable $N$ achieves the best performance on all tasks. Remarkably, its performance does not fluctuate much as $N$ changes from 5 to 40, often obtaining good and best results. We speculate that for simple tasks, either with small or big $\mathcal{M}$, the policy attention can always find optimal virtual policy $\psi$, and thus, ensures stable performance of MCPO. On the contrary, other methods observe a clear drop in performance as hyperparameters change (see Appendix B.2 for learning curves and the full table).

## 4.2 Navigation

Here, we validate our method on sparse reward environments using MiniGrid library [4]. In particular, we test MCPO and other baselines (same as above) on Unlock and UnlockPickup tasks. In these tasks, the agent navigates through rooms and picks up objects to complete the episode. The agent only receives reward +1 if it can complete the episode successfully. For sample efficiency test, we

| Model | HalfCheetah | Walker2d | Hopper | Ant | Humanoid | HumanoidStandup |
|---|---|---|---|---|---|---|
| TRPO | 2,811±114 | 3,966±56 | 3,159±72 | 2,438±402 | 4,576±106 | 145,143±3,702 |
| PPO | 4,753±1,614 | **5,278±594** | 2,968±1,002 | 3,421±534 | 3,375±1,684 | 155,494±6,663 |
| MDPO | 4,774±1,598 | 4,957±330 | 3,153±956 | 3,553±696 | 1,620±2,145 | 90,646±5,855 |
| TRGPPO | 2,811±114 | 5,009±391 | 3,713±275 | **4,796±837** | **6,242±1192** | 162,185±3755 |
| Mean $\psi$ | 4,942±3,095 | 5,056±842 | 3,430±259 | *4,570±548* | 353±27 | 71,308±11,113 |
| MCPO | **6,173±595** | *5,120±588* | ***3,620±252*** | 4673±249 | 4,848±711 | **195,404±32,801** |

Table 2: Mean and std. over 5 runs on 6 Mujoco tasks at 10M environment steps.

train all models on Unlock (find key and open the door) and UnlockPickup (find key, open the door and pickup an object), for only 100,000 and 1 million environment steps, respectively. The models use the best conservative hyperparameters found in the previous task (more in Appendix B.3).

Appendix's Fig. 3 shows the learning curves of examined models on these two tasks. For Unlock task, except for MCPO and VMPO, 100,000 steps seem insufficient for other models to learn useful policies. When trained with 1 million steps on UnlockPickup, the baselines can find better policies, yet still underperform MCPO. Here VMPO shows faster learning progress than MCPO at the beginning, however it fails to converge to the best solution. Our MCPO is the best performer, consistently ending up with average return of 0.9 (90% of episodes finished successfully). To achieve sample-efficiency, MCPO needs to store and search for good policies (rarely found in sparse reward problems), and adhere to it during optimization. MCPO's success may be attributed to utilizing the virtual policy that has the highest return.

To illustrate how the virtual policy supports MCPO's performance, we analyze the relationships between the old ($\theta_{old}$), the virtual policy ($\psi$) and the policies stored in $\mathcal{M}$ ($\theta_j$) throughout Unlock training. Fig. 1 (a) plots the location of these policies over a truncated period of training (from update step 5160 to 7070). Due to diversity-promoting rule, the steps where policies are added to $\mathcal{M}$ can be uneven (first row-red lines), often distributed right after the locations of the old policy (second row-green lines). We query at 10-th step behind the old policy (fourth row-cyan lines) to find which policy in $\mathcal{M}$ has the highest attention (third row-yellow lines, linked by blue arrows). As shown in Fig. 1 (a) (second and third row), the attended policy, which mostly resembles $\psi$, can be further or closer to the query step than the old policy depending on the training stage. Since we let the attention network learn to attend to the policy that maximizes the advantage of current mini-batch data, the attended one is not necessarily the old policy.

The choice of the chosen virtual policy being better than the old policy is shown in Fig. 1 (b) where we collect several checkpoints of virtual and old policies across training and evaluate each of them on 10 testing episodes. Here using $\psi$ to form the second KL constraint is beneficial as the new policy is generally pulled toward a better policy during training. That contributes to the excellent performance of MCPO compared to other single trust-region baselines, especially KL Fixed and Adaptive, which are very close to MCPO in term of objective function style.

## 4.3 Mujoco

Next, we examine MCPO and some trust-region methods from the literature that are known to have good performance on continuous control problems: TRPO, PPO, MDPO and TRGPPO (an improved version of PPO). To understand the role of the attention network in MCPO, we design a variant of MCPO ($N = 40$): *Mean $\psi$*, which simply constructs $\psi$ by taking average over policy parameters in $\mathcal{M}$. This baseline represents prior heuristic ways of building trust-region from past policies [28]. We pick 6 hard Mujoco tasks and train each model for 10 million environment steps. We report the results of best tuned models in Table 2 (see Appendix B.4 for full results).

The results show that MCPO consistently achieves good performance across 6 tasks, where it outperforms others significantly in HalfCheetah, Hopper, and HumanoidStandup. In other tasks, MCPO is the second best, only slightly earning less score than the best one in Walker2d and Ant. The variant Mean $\psi$ shows reasonable performance for the first 4 tasks, yet almost fails to learn on the last two. Thus, mean virtual policy can perform badly.

To understand the effectiveness of the attention network, we visualize the attention pattern of MCPO on the last two tasks and on Hopper-a task that Mean $\psi$ performs well on. Fig. 1 (c) illustrates that

| Model | PPO | ACKTR | VMPO | TRGPPO | MCPO |
|---|---|---|---|---|---|
| Mean | 131.19 | 195.52 | 18.20 | 116.80 | **229.99** |
| Median | 52.85 | 25.30 | 13.56 | 43.24 | **65.78** |

Table 3: Average normalized human score over 9 games. The performance of each run is measured by the best checkpoint during training over 10 million frames, averaged over 5 runs.

for the first two harder tasks, MCPO gradually learns to favor older policies in $\mathcal{M}$ ($j^* > 35$), which puts more restriction on the policy change as the model converges. This strategy seems critical for those tasks as the difference in average return between learned $\psi$ and Mean $\psi$ is huge in these cases. On the other hand, on Hopper, the top attended slots are just above the middle policies in $\mathcal{M}$ ($j^* \sim 25$), which means this task prefers an average restriction. As in most cases, MCPO with big policy memory ($N = 40$, more conservativeness) is beneficial. For those tasks where MCPO does not show clear advantage, we speculate that conservative updates are less important.

We also visualize $\alpha_t$ for HalfCheetah task, and observe an increasing trend across training steps (see Appendix Fig. 4). As the attention network gets trained, the virtual policy becomes better and can complement the old policy when the latter performs worse. Hence, on average, the weight $\alpha_t$ tends to be bigger overtime.

## 4.4 Atari Games

As showcasing the robustness of our method to high-dimensional inputs, we execute an experiment on a subset of Atari games wherein the states are screen images and the policy and value function approximator uses deep convolutional neural networks. We choose 9 typical games (6 were introduced in [14] and 3 randomly chosen) and benchmark MCPO against PPO, ACKTR, VMPO and PRGPPO, training all models for only 10 million environment steps. In this experiment, MCPO uses $N = 10$ and other baselines' hyperparameters are selected based on the original papers (see Appendix B.5).

As seen in Table 3, MCPO is significantly better than other baselines in terms of both mean and median normalized human score. The learning curves of all models are given in Appendix Fig. 5. To confirm MCPO maintains the leading performance with more training iterations, we train competitive models for 40 million frames for the first 6 games (Appendix Fig. 6). The results show that MCPO still outperforms other baselines after 40M frames.

## 4.5 Ablation Study

Finally, we verify MCPO's 3 components: virtual policy (in Eq. 3), switching-$\beta$ (Eq. 4) and diversity-promoting rule (Eq. 2). We also confirm the role of choosing the right memory size $N$ and learning to attend to the virtual policy $\psi$. In the task BipedalWalkerHardcore (OpenAI Gym), we train MCPO with different configurations for 50M steps. First, we tune $N$ (5,10 and 40) using the normal MCPO with all components on and find that $N = 10$ is the best. Keeping $N = 10$, we ablate or replace our component with an alternative and report the findings as follows.

**Virtual policy:** To show the benefit of pushing the new policy toward the virtual policy, we implement 3 variants of MCPO (N=10) that (i) does not use $\psi$'s KL term in Eq. 3 ($\alpha_t = 0$), (ii) use a fixed $\alpha_t = 0.5$ and (iii) only use $\psi$'s KL ($\alpha_t = 1.0$). All variants underperforms the normal MCPO by a margin of 100 or 50 return. **Switching-$\beta$:** The results show that compared to the annealed $\beta$ strategy adopted from MDPO and adaptive $\beta$ from PPO-KL, our switching-$\beta$ achieves significantly better results with about 50 and 200 return score higher, respectively. **Diversity-promoting writing:** We compare our proposal with the vanilla approach that adds a new policy to $\mathcal{M}$ at every update step (frequent writing) and other versions that write to $\mathcal{M}$ every interval of 10 and 100 update steps (uniform and sparse writing). Frequent, uniform and sparse writing all show slow learning progress, and ends up with low rewards. Perhaps, frequently adding policies to $\mathcal{M}$ makes the memory content similar, hastening the removal of older, yet maybe valuable policies. Uniform and sparse writing are better, yet it can still add similar policies to $\mathcal{M}$ and requires additional effort for tuning the writing interval. **Learned $\psi$:** To benchmark, we try alternatives: (1) using Mean $\psi$ and (2) only using half

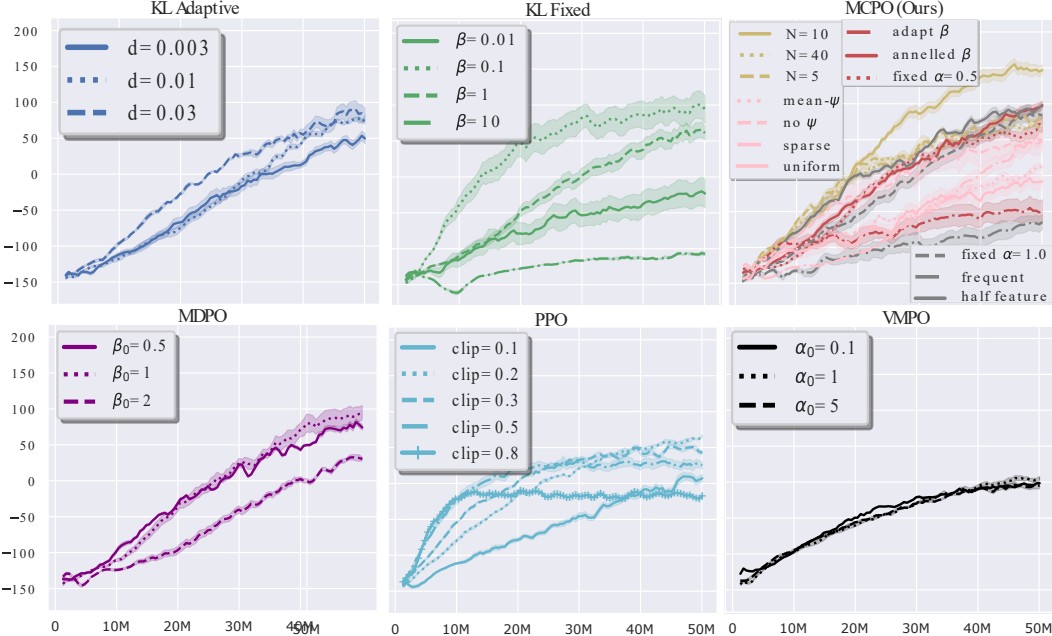

Figure 2: Ablation study on BibedalWalkerHardcore-v3: learning curves (mean and std. over 5 runs) across 50M training steps.

of the features in $v_{context}$ to generate $\psi$ (Eq. 1). The results confirm that the Mean $\psi$ is not a strong baseline for this task, only reaping moderate rewards. Using less features for the context causes information loss, hinders generations of useful $\psi$, and thus, underperforms the full-feature version by a margin of 50 return. For completeness, we also compare our methods to heavily tuned baselines and confirm that the normal MCPO ($N = 10$) is the best performer (see the details in Appendix B.6).

## 5 Related work

A framework for model-free reinforcement learning with policy gradient is approximate policy iteration (API), which alternates between estimating the advantage of the current policy, and updating the current policy's parameters to obtain a new policy by maximizing the expected advantage [2, 23]. Theoretical studies have shown that constraining policy updates is critical for API [6, 17, 20, 26]. An early attempt to improve API is Conservative Policy Iteration (CPI), which sets the new policy as a stochastic mixture of previous greedy policies [6, 16, 25]. Our paper differs from these works in three aspects: (1) we do not directly set the new policy to the mixture, rather, we use a mixture of previously found policies (the virtual policy) to define the trust region constraining the new policy via KL regularization; (2) our mixture can consist of more than 2 modes, and thus using multiple mixture weights (attention weights); (3) we use the attention network to learn these weights.

Also motivated by Kakade et al. (2002), TRPO extends the theory to general stochastic policies, rather than just mixture polices, ensuring monotonic improvement by combining maximizing the approximate expected advantage with minimizing the KL divergence between two consecutive policies [17]. Arguing that optimizing this way is too conservative and hard to tune, the authors reformulate the objective as a constrained optimization problem to solve it with conjugate gradient and line search. To simplify the implementation of TRPO, Schulman et al. (2017) introduces first-order optimization methods and code-level improvement, which results in PPO–an API method that optimizes a clipped surrogate objective using minibatch updates.

Constrained policy improvement can be seen as Expectation-Maximization algorithms where minimizing the KL-term corresponds to the Expectation step, which can be off-policy [1] or on-policy [22]. From the mirror descent perspective, several works also use KL divergence to regularize policy updates [31, 24, 20]. A recent analysis also points out the advantages of using KL term as a regu-

larizer over a hard constraint [7]. Some other works improve PPO with adaptive clip range [27] or off-policy data [5]. We instead advocate using 2 trust regions and apply our idea to a weaker backbone: PPO-penalty. Our method is also different from [5] as we do not use off-policy data/gradients to update the policy. Overall, our approach shares similarities with them where we also jointly optimize the approximate expected advantage and KL constraint terms for multiple epochs of minibatch updates. However, we propose a novel dynamic virtual policy to construct the second trust region as a supplement to the traditional trust region defined by the old or previous policy.

Prior works have promoted different utilization of memory to assist reinforcement learning. Unlike episodic memory concepts that stores observations across agent's life [3, 8], the memory buffer used in this paper stores the weight parameters of the policy network and attention mechanism is used to query this memory. The proposed memory instead can be viewed as an instance of program memory [10, 11]. That said, our memory is different from these prior works since our attention is learned through an auxiliary objective that optimizes the quality of the attended policy (program) with novel diversity-promoting writing mechanisms.

## 6    Discussion

We have presented Memory-Constrained Policy Optimization, a new method to regularize each policy update with two-trust regions with respect to one single old policy and another virtual policy representing multiple past policies. The new policy is encouraged to stay closer to the region surrounding the policy that performs better. The virtual policy is determined online through a learned attention to a memory of past policies. Compared to other trust-region optimizations, MCPO shows better performance in many environments without much hyperparameter tuning.

**Limitations**    Our method introduces several new components and hyperparameters such as $N$ and $\beta$. Due to compute limit, we have not tuned $\beta$ extensively and thus the reported results may not be the best performance that our model can achieve. Although we find the default values $\beta_{max} = 10$, $\beta_{min} = 0.01$ work well across all experiments in this paper, we recommend adjustments if users apply our method to novel domains.

**Negative Societal Impacts**    Our work aims to improve optimization in RL to reduce the sample complexity of RL algorithms. This aim is genuine, and we do not think there are immediate harmful consequences. However, we are aware of potential problems such as unsafe exploration committed by the agent (e.g. causing accidents) in real-world environments (e.g. self-driving cars). Finally, malicious users can misuse our method for unethical purposes, such as training harmful RL agents and robots. This issue is typical for any machine learning algorithm, and we will do our best to prevent it from our end.

## Acknowledgments

This research was partially funded by the Australian Government through the Australian Research Council (ARC). Prof Venkatesh is the recipient of an ARC Australian Laureate Fellowship (FL170100006).

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
