# Appendix

## A    Method Details

### A.1    The attention network

The attention network is implemented as a feedforward neural network with one hidden layer:

- Input layer: 12 units
- Hidden layer: $N$ units coupled with a dropout layer $p = 0.5$
- Output layer: $N$ units, $\mathrm{softmax}$ activation function

$N$ is the capacity of policy memory. The 12 features of the input $v_{context}$ is listed in Table 4.

Now we explain the motivation behind these feature design. From these three policies, we tried to extract all possible information. The information should be cheap to extract and dependent on the current data, so we prefer features extracted from the outputs of these policies (value, entropy, distance, return, etc.). Intuitively, the most important features should be the empirical returns, values associated with each policy and the distances, which gives a good hint of which virtual policy will yield high performance (e.g., a virtual policy that is closer to the policy that obtained high return and low value loss).

### A.2    The advantage function

In this paper, we use GAE [18] as the advantage function for all models and experiments

$$\hat{A}_t = \frac{1}{N_{actor}} \sum_i^{N_{actor}} \sum_{k=0}^{T-t-1} (\gamma\lambda)^k \left( V_{target} - V \left( s_{t+k}^i \right) \right)$$

where $\gamma$ is the discounted factor and $N_{actor}$ is the number of actors. $V_{target} = r_{t+k}^i + \gamma V \left( s_{t+k+1}^i \right)$. Note that Algo. 1 illustrates the procedure for 1 actor. In practice, we use $N_{actor}$ depending on the tasks.

### A.3    The objective function

Following [19], our objective function also includes value loss and entropy terms. This is applied to all of the baselines. For example, the complete objective function for MCPO reads

$$L = L^{MCPO} - c_1 \hat{\mathbb{E}}_t \left( V_\theta \left( s_t \right) - V_{target} \left( s_t \right) \right)^2$$
$$+ c_2 \hat{\mathbb{E}}_t \left[ -\log \left( \pi_\theta \left( \cdot | s_t \right) \right) \right]$$

where $c_1$ and $c_2$ are value and entropy coefficient hyperparameters, respectively. $V_\theta$ is the value network, also parameterized with $\theta$.

## B    Experimental Details

### B.1    Baselines and tasks

All baselines in this paper share the same setting of policy and value networks. Except for TRPO, all other baselines use minibatch training. The only difference is the objective function, which revolves around KL and advantage terms. We train all models with Adam optimizer. We summarize the policy and value network architecture in Table 5.

The baselines ACKTR, PPO[1], TRPO[2] use available public code (Apache or MIT License). They are Pytorch reimplementation of OpenAI's stable baselines, which can reproduce the original performance relatively well. For MDPO, we refer to the authors' source code[3] to reimplement the method.

---

[1] `https://github.com/ikostrikov/pytorch-a2c-ppo-acktr-gail`
[2] `https://github.com/ikostrikov/pytorch-trpo`
[3] `https://github.com/manantomar/Mirror-Descent-Policy-Optimization`

| Dimension | Feature | Meaning |
|---|---|---|
| 1 | $D\left(\theta, \psi_{old}\right)$ | "Distance" between $\theta$ and $\psi_{old}$ |
| 2 | $D\left(\theta_{old}, \psi_{old}\right)$ | "Distance" between $\theta_{old}$ and $\psi_{old}$ |
| 3 | $D\left(\theta_{old}, \theta\right)$ | "Distance" between $\theta_{old}$ and $\theta$ |
| 4 | $\hat{\mathbb{E}}_t\left[R_t\left(\psi_{old}\right)\right]$ | Approximate expected advantage of $\psi_{old}$ |
| 5 | $\hat{\mathbb{E}}_t\left[R_t\left(\theta_{old}\right)\right]$ | Approximate expected advantage of $\theta_{old}$ |
| 6 | $\hat{\mathbb{E}}_t\left[R_t\left(\theta\right)\right]$ | Approximate expected advantage of $\theta$ |
| 7 | $\hat{\mathbb{E}}_t\left[-\log\left(\pi_{\psi_{old}}\left(\cdot|s_t\right)\right)\right]$ | Approximate entropy of $\psi_{old}$ |
| 8 | $\hat{\mathbb{E}}_t\left[-\log\left(\pi_{\theta_{old}}\left(\cdot|s_t\right)\right)\right]$ | Approximate entropy of $\theta_{old}$ |
| 9 | $\hat{\mathbb{E}}_t\left[-\log\left(\pi_{\theta}\left(\cdot|s_t\right)\right)\right]$ | Approximate entropy of $\theta$ |
| 10 | $\hat{\mathbb{E}}_t\left(V_{\psi_{old}}\left(s_t\right) - V_{target}\left(s_t\right)\right)^2$ | Value loss of $\psi_{old}$ |
| 11 | $\hat{\mathbb{E}}_t\left(V_{\theta_{old}}\left(s_t\right) - V_{target}\left(s_t\right)\right)^2$ | Value loss of $\theta_{old}$ |
| 12 | $\hat{\mathbb{E}}_t\left(V_{\theta}\left(s_t\right) - V_{target}\left(s_t\right)\right)^2$ | Value loss of $\theta$ |

Table 4: Features of the context vector.

| Input type | Policy/Value networks |
|---|---|
| Vector | 2-layer feedforward net (tanh, h=64) |
| Image | 3-layer ReLU CNN with kernels $\{32/8/4, 64/4/2, 32/3/1\}$+2-layer feedforward net (ReLU, h=512) |

Table 5: Network architecture shared across baselines.

For VMPO, we refer to this open source code[4] to reimplement the method. We implement KL Fixed and KL Adaptive, using objective function defined in Sec. 2.

We use environments from Open AI gyms [5], which are public and using The MIT License. Mujoco environments use Mujoco software[6] (our license is academic lab). Table 6 lists all the environments.

## B.2 Details on Classical Control

For these tasks, all models share hyperparameters listed in Table 8. Besides, each method has its own set of additional hyperparameters. For example, PPO, KL Fixed and KL Adaptive have $\epsilon$, $\beta$ and $d_{targ}$, respectively. These hyperparameters directly control the conservativeness of the policy

---

[4] https://github.com/YYCAAA/V-MPO_Lunarlander
[5] https://gym.openai.com/envs/
[6] https://www.roboti.us/license.html

| Tasks | Continuous action | Gym category |
|---|---|---|
| Pendulum-v0 | X | Classical |
| LunarLander-v2 | | Box2d |
| BipedalWalker-v3 | ✓ | |
| Unlock-v0 | X | MiniGrid |
| UnlockPickup-v0 | | |
| MuJoCo tasks (v2): HalfCheetah Walker2d, Hopper, Ant Humanoid, HumanoidStandup | ✓ | MuJoCo |
| Atari games (NoFramskip-v4): Beamrider, Breakout Enduro, Gopher Seaquest, SpaceInvaders | X | Atari |
| BipedalWalkerHardcore-v3 | ✓ | Box2d |

Table 6: Tasks used in the paper.

| Model | Pendulum 1M | LunarLander 1M | BWalker 5M |
|---|---|---|---|
| KL Adaptive ($d_{targ} = 0.003$) | -407.74±484.16 | 238.30±34.07 | 206.99±5.34 |
| KL Adaptive ($d_{targ} = 0.01$) | -147.52±9.90 | *254.26±19.43* | 247.70±14.16 |
| KL Adaptive ($d_{targ} = 0.03$) | -601.09±273.18 | 246.93±12.57 | *259.80±6.33* |
| KL Fixed ($\beta = 0.01$) | -1051.14±158.81 | 247.61±19.79 | 221.55±38.64 |
| KL Fixed ($\beta = 0.1$) | -464.29±426.27 | *256.75±20.53* | *263.56±10.04* |
| KL Fixed ($\beta = 1$) | *-136.40±4.49* | 192.62±32.97 | 215.13±13.29 |
| PPO (clip $\epsilon = 0.1$) | -282.20±243.42 | 242.98±13.50 | 205.07±19.13 |
| PPO (clip $\epsilon = 0.2$) | -514.28±385.34 | *256.88±20.33* | 253.58±7.49 |
| PPO (clip $\epsilon = 0.3$) | -591.31±229.32 | *259.93±22.52* | *260.51±17.86* |
| MDPO ($\beta_0 = 0.5$) | *-136.45±8.21* | 247.96±4.74 | 251.18±29.10 |
| MDPO ($\beta_0 = 1$) | *-139.14±10.32* | 207.96±43.86 | 245.27±10.47 |
| MDPO ($\beta_0 = 2$) | *-135.52±5.28* | 227.76±16.96 | 226.80±15.67 |
| VMPO ($\alpha_0 = 0.1$) | *-144.51±7.04* | 201.87±29.48 | 236.57±10.62 |
| VMPO ($\alpha_0 = 1$) | *-139.50±5.54* | 212.85±43.35 | 238.82±11.11 |
| VMPO ($\alpha_0 = 5$) | *-296.48±213.06* | 222.13±35.55 | 164.40±40.36 |
| MCPO ($N = 5$) | **-133.42±4.53** | *262.23±12.47* | *265.80±5.55* |
| MCPO ($N = 10$) | -146.88±3.78 | *263.04±11.48* | **266.26±8.87** |
| MCPO ($N = 40$) | *-135.57±5.22* | **267.19±13.42** | 249.51±12.75 |

Table 7: Mean and std. over 5 runs on classical control tasks (with number of training environment steps). Bold denotes the best mean. Underline denotes good results (if exist), statistically indifferent from the best in terms of Cohen effect size less than 0.5.

update for each method. For MDPO, $\beta$ is automatically reduced overtime through an annealing process from 1 to 0 and thus should not be considered as a hyperparameter. However, we can still control the conservativeness if $\beta$ is annealed from a different value $\beta_0$ rather 1. We realize that tuning $\beta_0$ helped MDPO (Table 7). We quickly tried with several values $\beta_0$ ranging from 0.01 to 10 on Pendulum, and realize that only $\beta_0 \in \{0.5, 1, 2\}$ gave reasonable results. Thus, we only tuned MDPO with these $\beta_0$ in other tasks. For VMPO there are many other hyperparameters such as $\eta_0$, $\alpha_0$, $\epsilon_\eta$ and $\epsilon_\alpha$. Due to limited compute, we do not tune all of them. Rather, we only tune $\alpha_0$-the initial value of the Lagrange multiplier that scale the KL term in the objective function. We refer to the paper's and the code repository's default values of $\alpha_0$ to determine possible values $\alpha_0 \in \{0.1, 1, 5\}$. For our MCPO, we can tune several hyperparameters such as $N$, $\beta_{min}$, and $\beta_{max}$. However, for simplicity, we only tune $N \in \{5, 10, 40\}$ and fix $\beta_{min} = 0.01$ and $\beta_{max} = 10$.

On our machines using 1 GPU Tesla V100-SXM2, we measure the running time of MCPO with different $N$ compared to PPO on Pendulum task, which is reported in Table 9. As $N$ increases, the running speed of MCPO decreases. For this reason, we do not test with $N > 40$. However, we realize that with $N = 5$ or $N = 10$, MCPO only runs slightly slower than PPO. We also realize that the speed gap is even reduced when we increase the number of actors $N_{actor}$ as in other experiments. In terms of memory usage, maintaining a policy memory will definitely cost more. However, as our policy, value and attention networks are very simple. The maximum storage even for $N = 40$ is less than 5GB.

In addition to the configurations reported in Table 1, for KL Fixed and PPO, we also tested with extreme values $\beta = 10$ and $\epsilon \in \{0.5, 0.8\}$. Figs. 7, 8 and 9 visualize the learning curves of all configurations for all models.

## B.3 Details on MiniGrid Navigation

Based on the results from the above tasks, we pick the best signature hyperparameters for the models to use in this task as in Table 10. In particular, for each model, we rank the hyperparameters per task (higher rank is better), and choose the one that has the maximum total rank. For hyperparameters that share the same total rank, we prefer the middle value. The other hyperparameters for this task is listed in Table 8.

| Hyperparameter | Pendulum | LunarLander | BipedalWalker | MiniGrid | BipedalWaker Hardcore |
|---|---|---|---|---|---|
| Horizon $T$ | 2048 | 2048 | 2048 | 2048 | 2048 |
| Adam step size | $3 \times 10^{-4}$ | $3 \times 10^{-4}$ | $3 \times 10^{-4}$ | $3 \times 10^{-4}$ | $3 \times 10^{-4}$ |
| Num. epochs $K$ | 10 | 10 | 10 | 10 | 10 |
| Minibatch size $B$ | 64 | 64 | 64 | 64 | 64 |
| Discount $\gamma$ | 0.99 | 0.99 | 0.99 | 0.99 | 0.99 |
| GAE $\lambda$ | 0.95 | 0.95 | 0.95 | 0.95 | 0.95 |
| Num. actors $N_{actor}$ | 4 | 4 | 32 | 4 | 128 |
| Value coefficient $c_1$ | 0.5 | 0.5 | 0.5 | 0.5 | 0.5 |
| Entropy coefficient $c_2$ | 0 | 0 | 0 | 0 | 0 |

Table 8: Network architecture shared across baselines on Pendulum, LunarLander, BipedalWalker, MiniGrid and BipedalWaker Hardcore

| Model | Speed (env. steps/s) |
|---|---|
| MCPO (N=5) | 1,170 |
| MCPO (N=10) | 927 |
| MCPO (N=40) | 560 |
| PPO | 1,250 |

Table 9: Computing cost of MCPO and PPO on Pendulum.

### B.4 Details on Mujoco

For shared hyperparameters, we use the values suggested in the PPO's paper, except for the number of actors, which we increase to 16 for faster training as our models are trained for 10M environment steps (see Table 11).

For the signature hyperparameter of each method, we select some of the reasonable values. For PPO, the authors already examined with $\epsilon \in \{0.1, 0.2, 0.3\}$ on the same task and found 0.2 the best. This is somehow backed up in our previous experiments where we did not see major difference in performance between these values. Hence, seeking for other $\epsilon$ rather than the optimal $\epsilon = 0.2$, we ran our PPO implementation with $\epsilon \in \{0.2, 0.5, 0.8\}$. For TRPO, the authors only used the KL radius threshold $\delta = 0.01$, which may be already the optimal hyperparameter. Hence, we only tried $\delta \in \{0.005, 0.01\}$. The results showed that $\delta = 0.005$ always performed worse. For MCPO and Mean $\psi$, we only ran with extreme $N \in \{5, 40\}$. For MDPO, we still tested with $\beta_0 \in \{0.5, 1, 2\}$. Full learning curves with different hyperparameter are reported in Fig. 10. Learning curves including TRGPPO[7] are reported in Fig. 11

### B.5 Details on Atari

For shared hyperparameters, we use the values suggested in the PPO's paper, except for the number of actors, which we increase to 32 for faster training (see Table 11). For the signature hyperparameter of the baselines, we used the recommended value in the original papers. For MCPO, we use $N = 10$ to balance between running time and performance. Table 12 shows the values of these hyperparameters.

---

[7]We use the authors' source code `https://github.com/wangyuhuix/TRGPPO` using default configuration. Training setting is adjusted to follow the common setting as for other baselines (see Table 11).

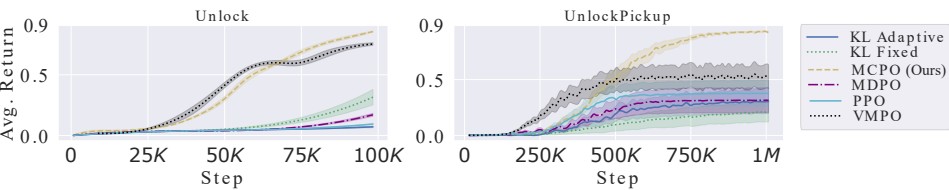

Figure 3: Unlock (left) and UnlockPickup (right)'s learning curves (mean and std. over 10 runs).

| Model | Chosen hyperparameter |
|-------|----------------------|
| KL Adaptive | $d_{targ} = 0.01$ |
| KL Fixed | $\beta = 0.1$ |
| PPO | $\epsilon = 0.2$ |
| MDPO | $\beta_0 = 0.5$ |
| VMPO | $\alpha_0 = 1$ |
| MCPO | $N = 10$ |

Table 10: Signature hyperparameters used in MiniGrid tasks.

| Hyperparameter | Mujoco | Atari |
|----------------|--------|-------|
| Horizon $T$ | 2048 | 128 |
| Adam step size | $3 \times 10^{-4}$ | $2.5 \times 10^{-4}$ |
| Num. epochs $K$ | 10 | 4 |
| Minibatch size $B$ | 32 | 32 |
| Discount $\gamma$ | 0.99 | 0.99 |
| GAE $\lambda$ | 0.95 | 0.95 |
| Num. actors $N_{actor}$ | 16 | 32 |
| Value coefficient $c_1$ | 0.5 | 1.0 |
| Entropy coefficient $c_2$ | 0 | 0.01 |

Table 11: Network architecture shared across baselines on Mujoco and Atari

We also report the average normalized human score (mean and median) of the models over 6 games in Table 3. As seen, MCPO is significantly better than other baselines in terms of both mean and median normalized human score. We also report full learning curves of models and normalized human score including TRGPPO in 9 games in Fig. 5 and Table 3, respectively.

Fig. 5 visualizes the learning curves of the models. Regardless of our regular tuning, VMPO performs poorly, indicating that this method is unsuitable or needs extensive tuning to work for low-sample training regime. ACKTR, works extremely well on certain games (Breakout and Seaquest), but shows mediocre results on others (Enduro, BeamRider), overall underperforming MCPO. PPO is always similar or inferior to MCPO on this testbed. Our MCPO always demonstrates competitive results, outperforming all other models in 4 games, especially on Enduro and Gopher, and showing comparable results with that of the best model in the other 2 games.

To verify whether MCPO can maintain its performance over longer training, we examine Atari training for 40 million frames. As shown in Fig. 6, MCPO is still the best performer in this training regime.

### B.6 Details on ablation study

In this section, we give more details on the ablated baselines. Unless state otherwise, the baseline use $N = 10$.

- **No $\psi$** We only changed the objective to

$$
\begin{aligned}
L_1 (\theta) = &\hat{\mathbb{E}}_t \left[ \tau_t (\theta) \hat{A}_t \right] \\
&- \beta \hat{\mathbb{E}}_t \left[ KL \left[ \pi_{\theta_{old}} (\cdot | s_t), \pi_\theta (\cdot | s_t) \right] \right]
\end{aligned}
\tag{7}
$$

| Model | Chosen hyperparameter |
|-------|----------------------|
| PPO | $\epsilon = 0.2$ |
| ACKTR | $\delta = 0.01$ |
| VMPO | $\alpha_0 = 5$ |
| MCPO | $N = 10$ |

Table 12: Signature hyperparameters used in Atari tasks.

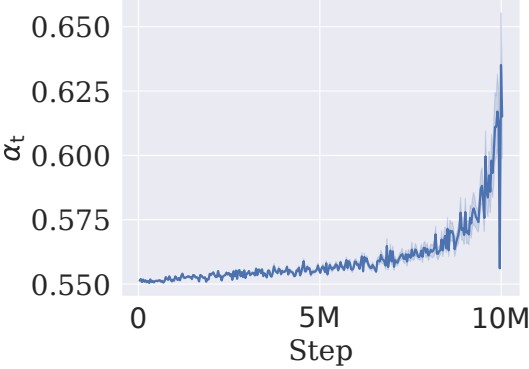

Figure 4: HalfCheetah: Average $\alpha_t$ over training time (mean and std. over 3 runs). The training does not use learning rate decay to ensure that $\psi$ and $\theta_{old}$ do not converge to the same policy towards the end of training.

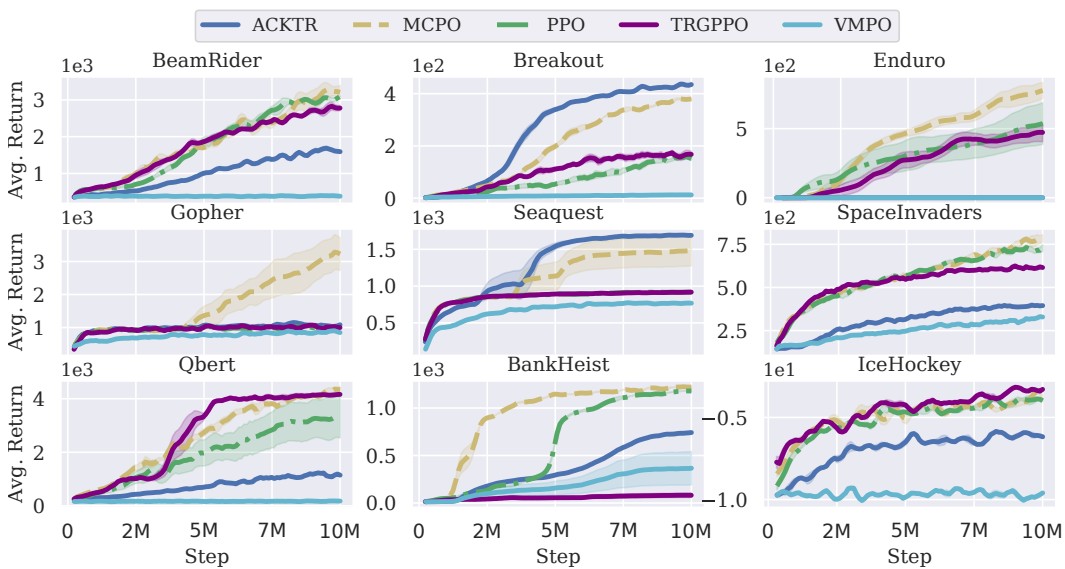

Figure 5: Atari games: learning curves (mean and std. over 5 runs) across 10M training steps.

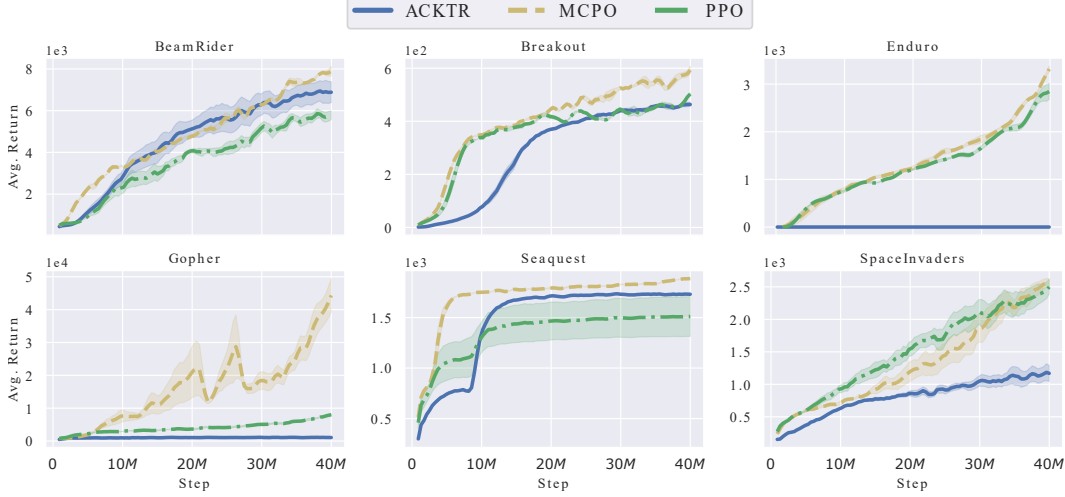

Figure 6: Atari games: learning curves (mean and std. over 5 runs) across 40M training steps.

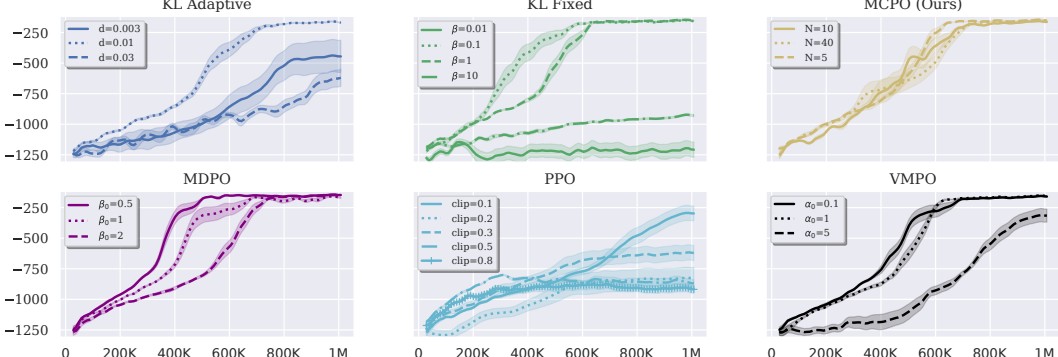

Figure 7: Pendulum-v0: learning curves (mean and std. over 5 runs) across 1M training steps.

where $\beta$ is still determined by the $\beta$-switching rule. This baseline corresponds to setting $\alpha = 0$

- **Fixed** $\alpha = 0.5$ We manually set $\alpha = 0.5$ across training. This baseline uses both old and virtual policy's trust regions but with fixed balanced coefficient.

- **Fixed** $\alpha = 1.0$ We manually set $\alpha = 1.0$ across training. This baseline only uses virtual policy's trust region.

- **Annealed** $\beta$ We determine the $\beta$ in Eq. 3 by MDPO's annealing rule, a.k.a, $\beta_i = 1.0 - \frac{i}{T_{total}}$ where $T_{total}$ is the total number of training policy update steps and $i$ is the current update step. We did not test with other rules such as fixed or adaptive $\beta$ as we realize that MDPO is often better than KL Fixed and KL Adaptive in our experiments, indicating that the annealed $\beta$ is a stronger baseline.

- **Adaptive** $\beta$ We adopt adaptive $\beta$, determined by the rule introduced in PPO paper (adaptive KL) with $d_{targ} = 0.03$.

- **Frequent writing** We add a new policy to $\mathcal{M}$ at every policy update step.

- **Uniform writing** Inspired by the uniform writing mechanism in Memory-Augmented Neural Networks [9], we add a new policy to $\mathcal{M}$ at every interval of 10 update steps. The interval size could be tuned to get better results but it would require additional effort, so we preferred our diversity-promoting writing over this one.

- **Sparse writing** Uniform writing with interval of 100 update steps.

- **Mean** $\psi$ The virtual policy is determined as

$$\psi = \sum_{j}^{|\mathcal{M}|} \theta_j \qquad (8)$$

- **Half feature** We only use features from 1 to 6 listed in Table 4.

The other baselines including KL Adaptive, KL Fixed, MDPO, PPO, and VMPO are the same as in B.2. The full learning curves of all models with different hyperparameters are plotted in Fig. 2.

## C  Theoretical analysis of MCPO

In this section, we explain the design of our objective function $L_1$ and $L_2$. We want to emphasize that the two trust regions (corresponding to $\theta_{old}$ and $\psi$) are both important for MCPO's convergence. Eq. 3 needs to include the old policy's trust region because, in theory, constraining policy updates using the last sampling policy's trust region guarantees monotonic improvement [17]. However, in practice, the old policy can be suboptimal and may not induce much improvement. This motivates us to employ an additional trust region to regulate the update in case the old policy's trust region is bad. In doing so, we still want to maintain the theoretical property of trust-region update while enabling a more robust optimization that works well even when the ideal setting for theoretical assurance does not hold.

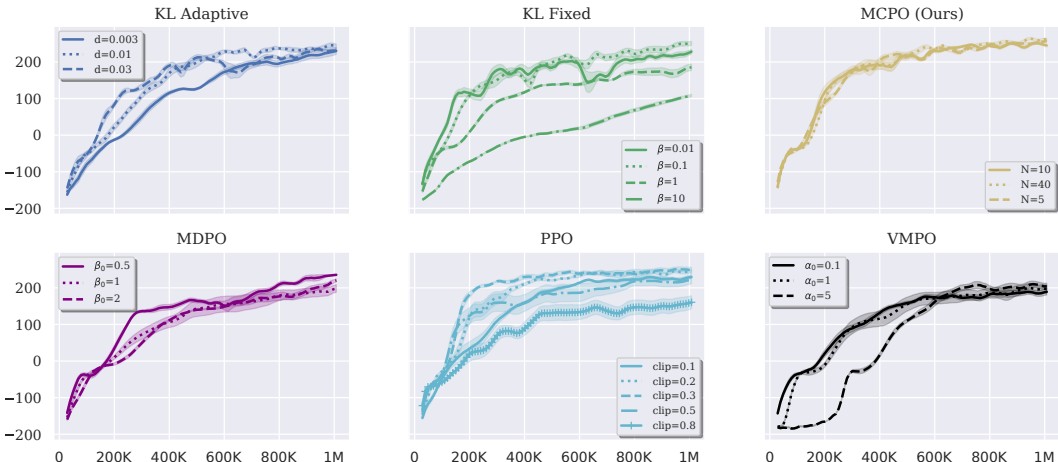

Figure 8: LunarLander-v2: learning curves (mean and std. over 5 runs) across 5M training steps.

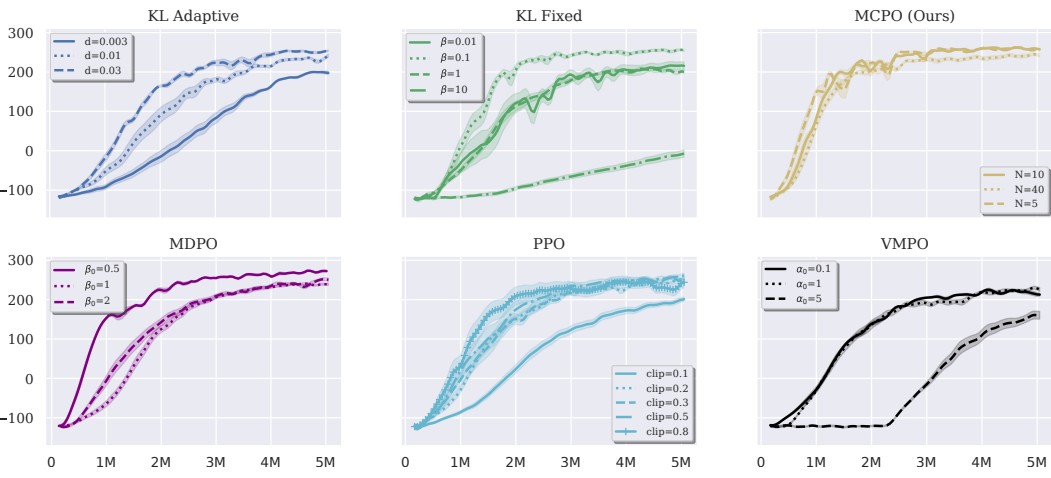

Figure 9: BipedalWalker-v3: learning curves (mean and std. over 5 runs) across 1M training steps.

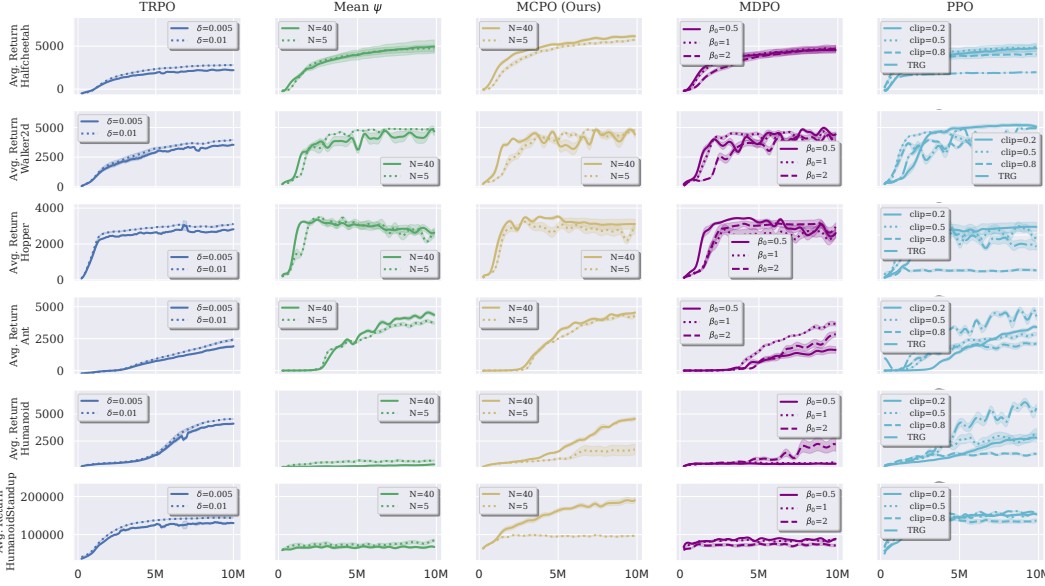

Figure 10: Mujoco: learning curves (mean and std. over 5 runs) across 10M training steps.

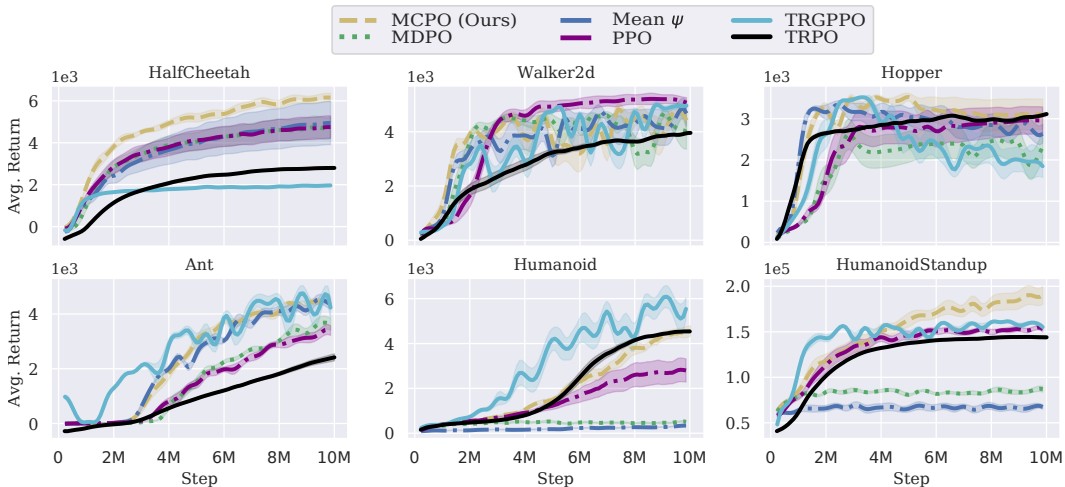

Figure 11: Mujoco: learning curves (mean and std. over 5 runs) across 10M training steps.

It should be noted that constraining with a policy different from the sampling one would break the monotonic improvement property of trust-region update. Fortunately, our theory proves that using the two trust regions as defined in our paper helps maintain the monotonic improvement property. This is an important result since if you use an arbitrary virtual policy to form the second trust region (unlike the one we suggest in this paper), the property may not hold.

Similar to [17], we can construct a theoretically guaranteed version of our practical objective functions that ensures monotonic policy improvement.

First, we explain the design of $L_1$ by recasting $L_1$ as

$$
\begin{aligned}
L_{1\theta_{old}}(\theta) = L_{\theta_{old}}(\theta) \\
- C_1 D_{KL}^{max}(\theta_{old}, \theta) \\
- C_2 D_{KL}^{max}(\psi, \theta)
\end{aligned}
$$

where $L_{\theta_{old}}(\theta) = \eta(\pi_{\theta_{old}}) + \sum_s \rho_{\pi_{\theta_{old}}}(s) \sum_a \pi_\theta(a|s) A_{\pi_{\theta_{old}}}(s,a)$–the local approximation to the expected discounted return $\eta(\theta)$, $D_{KL}^{max}(a,b) = \max_s KL[\pi_a(\cdot|s), \pi_b(\cdot|s)]$, $C_1 = \frac{4 \max_{s,a}|A_\pi(s,a)|\gamma}{(1-\gamma)^2}$ and $C_2 > 0$. Here, $\rho_{\pi_{\theta_{old}}}$ is the (unnormalized) discounted visitation frequencies induced by the policy $\pi_{\theta_{old}}$.

As the KL is non-negative, $L_{1\theta_{old}}(\theta) \leq L_{\theta_{old}}(\theta) - C_1 D_{KL}^{max}(\theta_{old}, \theta)$. According to [17], the RHS is a lower bound on $\eta(\theta)$, so $L_1$ is also a lower bound on $\eta(\theta)$ and thus, it is reasonable to maximize the practical $L_1$, which is an approximation of $L_{1\theta_{old}}$.

Next, we show that by optimizing both $L_1$ and $L_2$, we can interpret our algorithm as a monotonic policy improvement procedure. As such, we need to reformulate $L_2$ as

$$
L_{2\theta_{old}}(\psi) = L_{\theta_{old}}(\psi) - C_1 D_{KL}^{max}(\theta_{old}, \psi)
$$

Note that compared to the practical $L_2$ (as defined in the main paper on page 5), we have introduced here an additional $KL$ term, which means we need to find $\psi$ that is close to $\theta_{old}$ and maximizes the approximate advantage $L_{\theta_{old}}(\psi)$. As we maximize $L_{2\theta_{old}}(\psi)$, the maximizer $\psi$ satisfies

$$
L_{2\theta_{old}}(\psi) \geq L_{2\theta_{old}}(\theta_{old}) = L_{\theta_{old}}(\theta_{old})
$$

We also have

$$\eta\left(\theta\right) \geq L_{1\theta_{old}}\left(\theta\right) \qquad (9)$$
$$\eta\left(\theta_{old}\right) = L_{\theta_{old}}\left(\theta_{old}\right) \leq L_{2\theta_{old}}\left(\psi\right)$$
$$= L_{\theta_{old}}\left(\psi\right) - C_1 D_{KL}^{max}\left(\theta_{old}, \psi\right)$$
$$= L_{1\theta_{old}}\left(\psi\right) \qquad (10)$$

Subtracting both sides of Eq. 10 from Eq. 9 yields

$$\eta\left(\theta\right) - \eta\left(\theta_{old}\right) \geq L_{1\theta_{old}}\left(\theta\right) - L_{1\theta_{old}}\left(\psi\right)$$

Thus by maximizing $L_{1\theta_{old}}\left(\theta\right)$, we guarantee that the true objective $\eta\left(\theta\right)$ is non-decreasing.

Although the theory suggests that the optimal $L_2$ could be $L_2^* = \hat{\mathbb{E}}_t\left[R_t\left(\psi_\varphi\right) - C_1 KL\left[\pi_{\theta_{old}}\left(\cdot|s_t\right), \pi_\psi\left(\cdot|s_t\right)\right]\right]$, it would require additional tuning of $C_1$. More importantly, optimizing an objective in form of $L_2^*$ needs a very small step size, and could converge slowly. Hence, we simply discard the KL term and only optimize $L_2 = \hat{\mathbb{E}}_t\left[R_t\left(\psi_\varphi\right)\right]$ instead. Empirical results show that using this simplification, MCPO's learning curves still generally improve monotonically over training time.