# OpenReview forum: "Learning to Constrain Policy Optimization with Virtual Trust Region"
_NeurIPS.cc/2022/Conference — NeurIPS 2022 Accept_

### Official Review · Reviewer_gPBo · 2022-07-07

**Rating:** 5
**Confidence:** 4
**Soundness:** 2 fair
**Presentation:** 3 good
**Contribution:** 2 fair

**Summary:**

Most on-policy methods used a constraint only on the distance to the most current previous policy, but the constraint can deteriorate the performance of the current policy if the previous policy has worse performance than before. To tackle this issue, this paper proposes a new on-policy method called MCPO that uses two trust region constraints: constraint on 1) distance to the most current previous policy, and constraint on 2) distance to a good policy, called virtual policy, which is computed using previous policies stored in a policy memory. The parameters of the virtual policy are a weighted expectation of those of previous policies in the policy memory. The weights in the expectation are parameterized by a neural network called the attention network, whose input is a 12-dim feature vector, and the neural network is updated to maximize the estimated performance of the virtual policy. Empirical results show that the proposed MCPO outperforms other on-policy methods in most environments and show that all components of MCPO are essential to have the best performance through ablation studies.

**Questions:**

1. In the ablation study, are there more results of more variants of MCPO for each component?

2. Is there any reason for selecting BipedalWalkerHardcore for the experiments in the ablation study? Is there any result performed on other environments for ablation study?

**Limitations:**

While the authors give numerous experimental results to show the effectiveness of the proposed method, as aforementioned in weakness, it is still seen that it is a combination of simple and heuristic methods. In this case, I recommend two ways to the authors. The first is adding more results with various heuristic methods. In the ablation study, the authors considered only one simplest variant of the proposed one to compare with MCPO. To give reasons for why the chosen methods are reasonable, I think that the authors should measure the performance of at least 3 or more variants of MCPO for each component, and compare them with MCPO. The second way is to find the explanation of what the chosen methods mean through deriving equations. For example, considering the meaning of the term $(1-\alpha_t) KL[\pi_{old}, \pi_\theta] + \alpha_t KL[\pi_\psi, \pi_\theta]$ when $\alpha_t$ is the softmax of the true return of $\pi_\psi$ and $\pi_\theta$ can give a reasonable explanation for choosing the proposed components. If one of these is given, it helps readers understand the effectiveness of the chosen methods.

**Strengths And Weaknesses:**

Strength
1. The main ideas are 1) constraining both the previous policy and the virtual policy, 2) defining and optimizing the virtual policy using the attention network, and 3) constructing the 12-dim feature vector for input to the attention network, and these are simple but interesting.

2. Numerous empirical results show the proposed MCPO outperforms other on-policy algorithms in most environments and show the performance of variants of MCPO to ensure that all components of the method are essential to achieve the performance.


Weakness
1. Most of the components of the proposed method are designed heuristically based on some intuitions. For example, designing $\alpha_t$, switching $\beta$-rule, and the rule of storing a policy in the policy memory are all heuristically chosen methods. Although the authors showed the performance of MCPO with $\alpha_t = 0$, with annealing $\beta$, or with frequent writing and uniform writing, it is not enough to give reasons for why these methods improve the performance.

---

> ### Author Response · Authors · 2022-08-02
> **Reply to Reviewer gPBo**
>
> Thank you for your constructive comments. We answer your questions and concerns as follows.
>
> **Q1** Due to computation and time limits, we did not have chance to test all configurations. However, we think the number of experiments in our paper is significant, with each experiment often running at least five times. Therefore, in the ablation study, we could only include the typical configuration for each component. We also think that some components, such as $\alpha_t$, should be designed as in our paper to ensure the property of MCPO (preferring the trust-region of the better policy). The detailed meaning of $\alpha_t$ formulation is given in L147-150 (to answer your question in the Limitation). That said, we agree that more tests will be beneficial. We have run several additional ablation studies by adding the following baselines.
>
> 1. Regarding $\alpha$: MCPO (N=10) with fixed $\alpha=0.5$
>
> 2. Regarding $\alpha$: MCPO (N=10) with fixed $\alpha=1.0$
>
> 3. Regarding $\beta$: MCPO (N=10) with adaptive $\beta$, determined by the rule introduced in PPO paper (adaptive KL) with $d_{targ}$=0.03. The value is chosen based on the best result of Adaptive KL in Appendix Fig. 11
>
> 4. Regarding writing rule: MCPO (N=10) with uniform writing at every 100 update steps (sparse writing). This baseline is more conservative than the former variant (10 update steps)
>
> The test result after training is summarized in the table below.
>
> | Model       | Average Return (5 runs) |
> | ----------- | ----------------------- |
> | 1           | 91.83 $\pm$ 36.84       |
> | 2           | 114.74$\pm$28.36        |
> | 3           | -21.92$\pm$61.63        |
> | 4           | 73.11 $\pm$ 41.09       |
> | **MCPO (N=10)** | **169.24$\pm$29.61**      |
>
> The result shows that all other configurations are inferior to MCPO, confirming the choice of our component design. We have added the learning curves in the revision's  Appendix Fig.11 and will move it to the main manuscript when more space is allowed.
>
> **Q2** Reason for choosing BipedalWalkerHardcore:
>
> - It is a challenging environment. If we used simple settings such as those in Sec. 4.1, the performance gap between models would be negligible.
> - It is faster to run than other challenging tasks in Mujoco and Atari games. It is suitable for ablation study when we have to test many configurations.
> - It has many local optima, which helps discriminate better the level of performance of each configuration. Looking at Fig. 11 in the Appendix, we can see about six modes for the models to converge (unlike Mujoco ones which often have only 2-3 modes).
>
> Unfortunately, we currently do not have the compute resource to run all configurations in the ablation study for a new challenging task. Instead, we already reported a partial ablation study of $N$ and $\psi$ for classical control (Table 1) and Mujoco (Appendix Fig. 9).

---

> > ### Comment · Reviewer_gPBo · 2022-08-05
> > **Response to the authors**
> >
> > I thank the authors for their reply. I read the comments on my questions, but I still think that the proposed method is a combination of heuristic methods based on some intuitions. The additional results in Fig 11 show that the proposed method outperforms many variants of the proposed method. However, I think it would be better if the authors had explained why the combination of the chosen $\alpha_t$ and the chosen adjusting method for $\beta$ improves its performance through theory. I thank the authors again for their efforts to answer the questions I raised, but I will not change my score.

---

> > > ### Author Response · Authors · 2022-08-09
> > > **Regarding suggestion on theory**
> > >
> > > Thank you for your response. We are sorry that we could not satisfy the Reviewer despite running additional ablation studies for three configurations per component as requested by the Reviewer. While we think having a theory to explain the combination of our components is nice, we do *not* believe it is necessary to have such a theory to validate our model. First, each part of our model is well motivated and carefully designed to target specific issues. For example, $\alpha$ assigns the right weight to the old and virtual policy; $\beta$ leverages the policy memory to adaptively weight the trust region terms against the policy gradient term; and the *writing rule* ensures the policy memory meaningful, supporting the operations of other components.  Second, we provide empirical evidence to prove that each component is crafted adequately by comparing our design with three alternatives, as you suggested. The results demonstrate that our components work as expected. The theory is nice to have and worth further investigation in future work. However, for the scope of this work, we do not think it should be a compulsory element to make our work a good paper.

---

### Official Review · Reviewer_HVYg · 2022-07-09

**Rating:** 3
**Confidence:** 5
**Soundness:** 1 poor
**Presentation:** 2 fair
**Contribution:** 1 poor

**Summary:**

The paper considers the trust-region-based policy optimization methods in deep reinforcement learning for which the policy update (from an old policy to a new one) at each iteration should be constrained by a trust region. This paper argues that for this family of methods the trust region constraint could impede the policy update since the old policy might not necessarily be a good one to define the trust region. The paper then proposes an improvement by incorporating into the TRPO objective a second type of trust region, which is defined as the KL divergence between the target policy and a linear combination of some history policies, i.e., *virtual trust region*. In order to effectively find such linear combination, the paper presents an attention-based mechanism to learn the combination coefficients based on some contextual features of the policy. The paper also empirically evaluates the augmented trust region method, Memory-Constrained Policy Optimization (MCPO) on both discrete and continuous control tasks.

**Questions:**

Q1: After digging into Eq. (3), I found the virtual trust region could be replaced by the cross entropy between $\pi_\psi$ and $\pi_\theta$ without affecting the maximization. In this sense, having the “virtual trust region” could be effectively imposing an entropy regularization. Could it be possible that having an entropy regularization (i.e., maximum entropy) for TRPO and all other trust-region based methods would yield a similar empirical performance as MCPO? We can try decaying entropy coefficient or tuning the coefficient based on the learning progress.


Q2: (line 151) How to estimate $R_t(\psi)$ from $A_t(\theta_{old})$ via importance sampling given that their state distribution are also different?

Q3: (algo 1 line 6) How to compute $\psi$ via Eq. 1 using $\psi_{old}$, $\theta$, and $\theta_{old}$?

Q4: How the parameter $\varphi$ is updated when optimizing Eq. (6)?

Q5: (line 149) Why the $\alpha_t$ is defined as the ratio of return exponents? This $\alpha_t$ is different from what’s being used in Eq. 3 as it does not condition on $s_t$.

Q6: (line 255 to 256) the paper states that “conservative updates are less important in some tasks”. Does it mean the virtual trust region constraint could impede training in some environments? This would be contradictory to the underlying motivation of this paper.

**Limitations:**

yes

**Strengths And Weaknesses:**

**Strengths**:

[clarity] The paper presents an interesting idea of “extrapolating” the trust region by introducing a virtual one. The way of learning an attention-based mechanism to determine the trust region is also inspiring.


**Weaknesses**:

[originality] Lack of convincing evidence to motivate the work and method: The paper argues that the trust region defined according to the old policy could lead to “suboptimal policy update” (line 34 to 36, line 103 to 104). This argument is interesting. It would be more convincing if the paper could provide any empirical evidence to support such argument. Also, it is unclear how the “new policy may fall into a local optimum”. The trust region constraint itself is to ensure that updating policy could result in a performance improvement. That is, as long as the policy can be improved, regardless how much the improvement is, the trust region is then effective and the policy update is valid. Thus, it remains unclear to me how such local optimum would really impact the training.

[clarity & significance] Lack of important technical details: One of major contributions of this paper is introducing “virtual trust region”, which is defined as the KL divergence between the psi policy and the new policy. The psi policy is constructed based on the context vector (Tab 4). However, it remains largely unknown what the distance between $\theta$ and $\psi$ is (feature dim 1, 2 & 3), how the return of a new policy (before it is updated) is computed (feature dim 6, 10, 11 & 12). Furthermore, it is also unclear how the psi weights are optimized. It is not straightforward how Eq. (5) is maximized w.r.t. $\varphi$. The lack of these important details could downplay the significance of this work.

[experimental results] Lack of insightful results to support how virtual trust region works: The paper presents many empirical results of MCPO. Most of them are from the lens of a whole policy training process, e.g., empirical returns. It is, however, largely unclear what role the virtual trust region plays at each policy iteration. Given that the main idea of the paper is about how a “bad” old policy would affect the trust region constraint, it would make more sense to dig into the trust region analysis (e.g., empirical KL estimates) at each iteration, rather than the empirical returns (one such analysis can be found in PPO paper, Figure 2). Moreover, the ablation studies presented in the current version are very inconclusive. No detailed training curves or analytical statistics are available.

---

> ### Author Response · Authors · 2022-08-02
> **Reply to Reviewer HVYg (part 1)**
>
> Thank you for your thoughtful review. Please see below our response to your questions and concerns.
>
> ### Weaknesses:
>
>  **[Originality]**. In theory, trust-region methods guarantee monotonic improvement. However, the theory often makes unrealistic assumptions. For example, in TRPO-one of the most rigorous theory-guaranteed paper, Theorem 1 requires the true advantage function $A$, which is computed from the true state-action value function. In practice, we use a neural network to estimate the value function, so the theory may not hold. Also, in practice, TRPO uses the average KL divergence instead of the $D_{KL}^{max}$ required by the theory, not to mention the optimization error when solving the constrained optimization problem (Eq. 13 in TRPO paper). Other methods such as PPO are not even theoretically guaranteed. Therefore, in practice, it is possible that using one trust region from the old policy can: (1) jump into a local optimum with worse performance and (2) get stuck in the local optimum for a while.
>
> We can find many empirical pieces of evidence in our paper and others to support (1) and (2). In the TRPO paper, the learning curves in Fig. 4 (despite being averaged) can be up and down. This means the policy sometimes jumps to deficient regions that break the monotonic improvement. The same happens in the PPO paper (Fig. 3), demonstrating the issue (1). For point (2), as shown in our paper (Appendix Fig 10), in complex tasks such as HalfCheetah or HumanoidStandup, many trust-region baselines (including TRPO and PPO) converge to a local optimum, showing almost horizontal learning curves until the training finishes.
>
> **[clarity & significance]** Due to space constraints, we have to compress the text and move some technical details to the Appendix, and thus can confuse you.  We believe that we can fix the problem when more space is allowed. Below we address your questions.
>
> **"unknow what the distance ..."**. In L127, we defined $D$ as the KL divergence denoting the "distance".
>
> **"how the return ..."**. The return was defined in L151. The advantage is collected at the beginning of the update phase, as shown in Algo. 1.
>
> **"how the psi weights are optimized."**. As explained in L158-161, we optimize the attention network $\varphi$, not $\psi$. The RHS of Eq. (5) is a function of the attention weight, and the attention weight is a function of $\varphi$. The gradient $\frac{\partial L_2}{\partial \varphi}$ can be backpropagated as usual. We have clarified it in the revision.
>
> **[experimental results]**. Thank you for your suggestion on trust region analysis. Fig. 2 in the PPO paper compares surrogate functions to show that $L^{clip}$ behaves as expected (lower-bound to $L^{CPI}$ and reduces as the KL increases too much). We expect different behaviour from MCPO and thus report another type of analysis. We want to confirm that MCPO can prefer $\psi$ over $\theta_{old}$ (see Appendix Fig. 3) as the performance of $\psi$ is better than that of $\theta_{old}$ (Fig. 1b). We also demonstrated that during optimization, the attention network actually uses different policies rather than $\theta_{old}$ (Fig. 1a), which indicates that the performance gain of MCPO is not attributed to using the traditional trust-region surrounding $\theta_{old}$.
>
> **"No detailed training curves .."** As mentioned in L298, we provided the details in Appendix B.6, showing the learning curves of all models (Appendix Fig. 11).

---

> ### Author Response · Authors · 2022-08-02
> **Reply to Reviewer HVYg (part 2)**
>
> ### Questions:
>
> **Q1**. Yes, you can interpret it that way. We note that we are enforcing $\pi(\theta)$ to close to $\pi(\psi)$, not merely maximizing the entropy of $\pi(\theta)$. Recent methods such as PPO already include entropy maximization in the objective function. Our MCPO implementation also adopts this, as shown in Appendix A.3. We did not tune/decay the coefficient and used the default value suggested in the PPO paper, which the authors already tuned. In short, our MCPO implementation is the same as PPO's, except that we replace $L^{clip}$ with our $L^{MCPO}$. Therefore, the difference in performance must attribute to $L^{MCPO}$, not the entropy maximization.
>
> **Q2&Q5**. There seems to be a misunderstanding. $R_t(\psi)$ and $A_t(\theta_{old})$ condition on the same state $s_t$. We have made it clear in the revision (actually in L73, we defined $A_t$ as a function of ($s_t$ and $a_t$)). $\alpha_t$ in L149 is the same as in Eq. (3). We missed notation $(|s_t)$ and have fixed it in the revision (thank you for noticing it). The motivation for $\alpha_t$ is given in L147-148. Intuitively, if $\psi$ is better, its return is higher than $\theta_{old}$, and so $\alpha_t$. Then, Eq. (3) will lay more emphasis on the virtual policy's trust region.
>
> **Q3**. As explained in Appendix A.1 and Table 4, we use $\theta_{old}$, $\psi_{old}$ and $\theta$ to compute the feature $v$. The attention network takes $v$ as input to compute $\psi$ as in Eq. (1).
>
> **Q4**. As explained in response to your "weakness" section, we use gradient ascent and let the gradient backpropagate to update $\varphi$'s parameters.
>
> **Q6**. We did not mean it could impede the training. We meant it might provide not much advantage against the case that requires conservative updates. For example, in a simple task like Hopper, where almost all policies can lie in the optimal region of policy space, the virtual trust region may not help much. MCPO still shows competitive performance in these cases. However, there is no such big performance gain as in HalfCheetah and HumanoidStandup.

---

### Official Review · Reviewer_ECcM · 2022-07-11

**Rating:** 6
**Confidence:** 4
**Soundness:** 3 good
**Presentation:** 3 good
**Contribution:** 3 good

**Summary:**

This work addresses the problem of policy optimization in policy gradient methods (PG) in deep RL. Due to the off-policy nature of using mini-batch data to make updates to the policy, the policy updates can become very far from the areas where data has been collected. In this work, the authors propose to construct a so-called virtual policy that is a weighted mixture of several past policies, where the weights are learned via an attention network. The policy updates are then regularized to stay within a KL-distance region around the virtual policy and the previous behaviour or “old policy”. In this way, if the old policy performs poorly or is stuck in a local optimum, the new policy will be encouraged towards the virtual policy. The authors provide empirical evidence of their proposed method having better performance than some previous works that also form trust regions or constraints of similar kinds (i.e. TRPO, PPO, and the like) on a variety of domains.

**Questions:**

- One potential issue I see with the way Eqn (2) chooses which new policy to include in its memory is that if the list of policies becomes very uniform $D(\theta_{old}, \psi)$ would become very small and many policies would get added, potentially making the usage of the memory useless. Have the authors observed something like this in their experimentation and how do they address it?
- L253-254: Regarding Hopper “prefers an average restriction”. Not sure I understand this point. What does it mean that it prefers an average restriction? To my understanding, since MCPO favours better-performing policies, this means that the ~middle policy performed the best, but on average all the policies were not very bad since "Mean $\psi$" also performed similarly. Are the authors saying this is related to the environment? If so, what kind of qualities of the environment might lead to this? Or is this more related to the optimization procedure?
- How do we know if the regularization is working, i.e. the new policy is closer to $\psi$ when the old policy becomes stuck or bad, and that this is the reason for improved results? I think some hand-designed experiments that demonstrate or visualize the effect of having the virtual policy would be interesting (similar to Figure 1 but more comprehensive, i.e. including situations where $\theta_{old}$ is forced to be a bad or locally optimum policy)


**Limitations:**

The authors discuss some reasonable limitations in their work.

**Strengths And Weaknesses:**

The proposed method of keeping a virtual policy that keeps track of past policies in order to avoid the pitfalls of simply keeping close to a potentially bad or suboptimal old policy is straightforward and sensible. I also think the use of a learned attention network to weigh these policies is a good idea. The empirical evaluation also shows good results on a wide variety of domains.
I think the work lacks in a theoretical understanding of the impact of the proposed method. Does the virtual policy really help escape local optima? Are the improvements observed in performance really due to the virtual policy or are they due to for example, an improved optimization with the new objective?

Strengths:
- Good idea, simple to implement and use, asymptotic results, and learning curves (in the appendix) appear to consistently show improved performance over baselines.

Weaknesses:
- In-depth understanding of the impact of virtual policy as stated above.

Writing:
- Overall, the writing quality is good, and the paper is easy to follow.
- Definitions of setting is missing, for example I did not realize we were in a finite-horizon setting until I read Alg 1.
- L168: typo: optimize $\theta$ and *$\phi$* alternately
- L178: N or M (L113-115) for number of past policies?
- Section 4.2 results are in the Appendix but talked about in the main body. I understand space constraints but I would recommend only talking in-depth about results that can be shown in the main body for improved readability.

---

> ### Author Response · Authors · 2022-08-02
> **Reply to Reviewer ECcM**
>
> We thank the reviewer for your insightful feedback. We address your questions in the following.
>
> ### Weaknesses:
>
> **"lacks in a theoretical understanding ..."**. We can prove that under certain assumptions (perfect value estimation, well-trained attention network, etc.), MCPO with virtual policy guarantees monotonic improvement (see Appendix C). In practice, proving that an algorithm can overcome local optima (due to imperfect value estimation and other approximations, etc.) is challenging. That said, we agree that having a detailed theoretical analysis of the property of the virtual policy would be valuable. We compensate for the lack of such theoretical analysis with empirical results that show the virtual policy's benefits, as presented below to answer your questions.
>
> **"Does the virtual policy really help..."**  In the ablation study (see Appendix Fig. 11 for clear comparison), the performance of MCPO without the virtual policy $\psi$ (No $\psi$) converges to a local optimum, resulting in ~50 scores. In contrast, MCPO with $\psi$ can get around this local optimum after around 20M steps and finally achieves ~150 scores. Similarly, for complex tasks such as Half-Cheetah and HumanoidStandup where local optima impede the performance of other methods such as PPO (see Appendix Fig. 10), MCPO can break the bottleneck and reach much better results. The main difference between MCPO and PPO is the KL constraint regarding the virtual policy (Eq. 1). Therefore, the virtual policy must constitute that change in learning behaviour.
>
> ### Writing:
>
> **"Definitions of setting ..".**. Thank you for pointing it out. We have made this point early in the revision.
>
> **L168**. Thank you, we have fixed it in the revision.
>
> **L178**. It is $N$, as defined in Line 124
>
> **Section 4.2 results**. Thank you for being so understanding. We will move Fig. 11's MCPO to the main text when more space is allowed.
>
> ### Questions:
>
> **"One potential issue ..."**. This is an interesting finding. Even when we assume that the list of policies is very uniform and many new policies are added, this only happens for a short time. Since it is unlikely that all the newly added policies are similar to those in the list, these new policies will make the list less uniform, and thus enable Eq. (2) to work correctly. This phenomenon is illustrated in Fig. 1 (a)'s red vertical lines. At the beginning of the update phase, many policies enter the list. Yet, later, only some can pass Eq. (2). Increasing $N$ will help alleviate your concern further since it reduces the chance of uniformity, which partially explains why larger $N$ often yields better performance in challenging experiments (see Appendix Fig. 9).
>
> **L253-254**. Your understanding is correct. Here, the behaviour depends on the task difficulty, which may positively correlate with the number and the depth of local optima in the policy space. For example, Half-Cheetah has a well-known local optimum where the agent "walks" upside down. Many RL algorithms fall into this trap and earn low scores. For Hopper, it is easier to get over local optima, and the performance of MCPO is already very close to the task's reward threshold (3800), which means at the end of the training, almost all policies lie in the optimal region of policy space. Hence Mean-$\psi$ performance is not bad.
>
> **"How do we know ..."**. We can keep track of the performance of $\theta_{old}$ and $\psi$ (like Fig. 1 (b)) and the coefficient $\alpha$ (like Appendix Fig. 3). We often observe that when the average test return of $\psi$ start becoming better than that of $\theta_{old}$, $\alpha$ also starts increasing, which mean the algorithm is forcing the new policy to get closer to $\psi$.

---

> > ### Comment · Reviewer_ECcM · 2022-08-09
> > **Response to authors**
> >
> > Thank you for your detailed response to my comments. After reading them as well as the discussions with the other reviewers, I still feel that a more technical understanding of why the method works or should work is needed, which I believe is also noted by the other reviewers. The problem is that although the intuitions make sense, and there are some empirical evidence supporting them, they are not enough to convince me that the intuitions are sound.
> >
> > If the authors could demonstrate theoretically through some simple examples or domains or empirically through simple domains as illustrations that their claims are validated, that would greatly strengthen the work.
> >
> > I still think the empirical evidence shows promise however, and so I will keep my current score.

---

### Author Response · Authors · 2022-08-02
**General response**

Dear Reviewers,

We appreciate your effort in reviewing our paper. We are glad that the reviewers find: (1) our idea "good" (Reviewer ECcM), "interesting" (Reviewer HVYg and gPBo) and "inspiring" (Reviewer HVYg); (2) our experiments "consistently show improved performance" (Reviewer ECcM) and having "numerous empirical results" (Reviewer gPBo); and (3) our writing "good" and "easy to follow" (Reviewer ECcM). However, there remain misunderstandings and questions needed clarification. We will address these by replying to each of your reviews. We hope that our responses will address your concerns. Please consider increasing your score if you find our responses valid.

---

### Author Response · Authors · 2022-08-03
**Revision summary**

Dear Reviewers,

We summarize the change in the current revision:
- Fix typos
- Add more descriptions to clarify the concerns as mentioned in individual replies
- Add more ablation studies as suggested by Reviewer gPBo

These changes may make the line number different from the last version. Therefore, please refer to the submitted version if you want to verify our explanation mentioning the line numbers in our responses.
We would be glad to further clarify any points in the upcoming reviewer-author interaction week.

---

### Meta-Review · Area_Chair_aqYX · 2022-08-28

**Recommendation:** Accept
**Confidence:** Less certain

**Metareview:**

The paper addresses a constrained policy optimization through an addition of a virtual trust region trained with an attention. Evaluations show that the proposed method outperforms other on-policy methods most of the time.

The reviewers agree that the method is novel and effective, and the evaluations are extensive. The revised paper added ablation studies that show effects of different components of the method, although some reviewers would like to see deeper analysis.

**Award:**

No

---

### Decision · Program_Chairs · 2022-09-14

Accept